# Financial and Economic Investment Evaluation of Wastewater Treatment Plant

**Jasmina Ćetković [1], Miloš Knežević [2], Slobodan Lakić [1], Miloš Žarković [1,*] , Radoje Vujadinović [3] , Angelina Živković [4] and Jelena Cvijović [5]**

1   Faculty of Economics Podgorica, University of Montenegro, 81000 Podgorica, Montenegro; cetkovicjasmina2@gmail.com (J.Ć.); sasalakic@mail.com (S.L.)
2   Faculty of Civil Engineering, University of Montenegro, 81000 Podgorica, Montenegro; knezevicmilos@hotmail.com
3   Faculty of Mechanical Engineering, University of Montenegro, 81000 Podgorica, Montenegro; radojev@ucg.ac.me
4   Independent Researcher, 81000 Podgorica, Montenegro; angelina.zivkovic.01@gmail.com
5   Faculty of Organizational Sciences, University of Belgrade, 11000 Belgrade, Serbia; vuksanovicjelena1@gmail.com
*   Correspondence: milos.zarkovic87@gmail.com; Tel.: +382-67-569-033

**Abstract:** Improved Cost-Benefit Analysis (CBA) analysis requires a broader analytical framework, in order to perceive each project individually from the perspective of potentially measurable and significant effects on the environment and society as a whole. The main goal of our paper is to assess the financial and economic justification for variant V3 (as the most technically optimal) of the wastewater treatment plant (WWTP) construction project in Nov Dojran, North Macedonia, with the purpose of advancing municipal infrastructure and environmental benefits from improved water treatment. Based on the economic analysis conducted, we conclude that the investment in the WWTP project is justified, because the economic internal rate of return is higher than the opportunity cost of capital (EIRR = 16.38%), the economic net present value is higher than 0, and EBCR (benefit-cost ratio) is greater than 1 (EBCR = 2.11). The highest environmental benefit of 49.2% in total environmental benefits is associated with nitrogen, while phosphorus is the next pollutant in the structure of environmental benefits at 46.1%. The environmental benefits of removing biological oxygen demand (BOD) and chemical oxygen demand (COD) are significantly less important, despite the removal of significant amounts of these pollutants during treatment. The situation is similar with suspended particles.

**Keywords:** economic analysis; wastewater treatment; pollutants; environmental benefits





## 1. Introduction

The development of social awareness in the balance of human development and environmental protection is a prerequisite for the concept of sustainable development. Sustainable development policies and strategies should be based on greenways (GWs) [1], which represent the potential for an integrated strategy of simultaneous environmental quality and economic development [2,3]. Recent research suggests that certain sectors positively stimulate environmental degradation, which highlights the importance of institutional excellence in combating degradation [4,5]. Even certain long-term assessments have shown that financial development significantly enhances environmental degradation, which should be urgently considered by policymakers who, within sustainable environmental policy and green financing approaches, should seek solutions for this problem [6–8].

Intensified urban and industrial development, population growth, as well as the actualization of climate change, are putting serious pressure on local water resources. On the other hand, the scientific community has intensified efforts towards the protection of

water resources [9,10], which indicates significant problems in this area. The fact is that Europe has long faced water stress, not only in terms of lack of this key resource but also in terms of problems with deteriorating water quality and adequate wastewater treatment. In this regard, in the last few decades, the concept of sustainable wastewater management has been strongly articulated [11]. The concept itself is multidimensional as it implies ecological, socio-cultural, and economic sustainability [12]. There are many definitions of wastewater in the literature, but according to a widely accepted definition, wastewater is a combination of household waste (consisting of blackwater and greywater), water from commercial facilities and institutions (including hospitals), industrial wastewater, atmospheric wastewater, and wastewater from agriculture, horticulture, and aquaculture [13].

The growing demand for water resources, on the one hand, and the pronounced degradation of ecosystems, on the other hand, is a complex challenge for policymakers in how to establish the adequate ecological status of water bodies and implement sustainable wastewater treatment. The EU Directive concerning urban wastewater treatment from 1991 obliged all generated wastewater agglomerations (between two and ten thousand equivalent inhabitants) to establish adequate systems for wastewater collection and treatment (until December 2005). Thus, the need for upgrading the applied treatment was articulated, as well as the construction of new facilities for wastewater treatment.

However, recent research continues to point to a problem within certain industries which, despite being among the heaviest polluting industries, still operate in a conventional way and are not focused on preventing water pollution [14].

As water pollution by biological and chemical contaminants is a significant problem for industries, for decision makers, and society in general, understanding the advantages and disadvantages of available technologies for wastewater treatment is just one way to keep this problem under control [15]. The application of a multicriteria decision-making theory can help stakeholders in selecting a particular type of technology whose application is the best and most justified under given conditions. Modern technologies for wastewater reduction and treatment [16–18], as well as wastewater reuse [19–22], are major tools for conserving usable water.

There are a lot of problems in the field of wastewater management, and we will only state some of them. Conventional wastewater treatment plants (WWTPs) reduce water pollution but contribute to air pollution because their inputs (energy and materials) are linked to emissions [23,24]. Therefore, LCA (life cycle assessments) are increasingly used to assess the impact of these plants on pollution, all with the aim of applying such WWTPs that have zero impact on the environment [25,26]. Furthermore, recent research offers solutions for key energy savings of WWTPs to provide energy and financial savings during wastewater treatment. In order to meet the concept of sustainable wastewater treatment (which does not generate secondary wastewater pollution and does not require external energy), efforts are being made to research modern wastewater treatment technologies which fulfill these expectations. An alternative to conventional power WWTPs can be solar energy, especially in regions with a large number of sunny days per year [27].

In order to reduce operating costs and environmental damage from WWTPs, one study points to a model developed for compressing air storage in WWTPs, which can be designed for specific WWPTs depending on plant capacity and wastewater characteristics. Anaerobic digestion (AD) from different organic sources can be used, from which fully renewable biogas with high energy value can be provided. Another solution is the co-digestion of sewage sludge with other organic substrates to increase the methane obtained from AD. At the same time, the upgrade of biogas with high energy value biomethane is technically comparable to fossil-derived natural gas from fossil fuels [28]. Campana et al. developed a highly flexible and widely applicable generalization model, with different scales of WWTPs. The mathematical model of simulation and optimization enables the connection of renewable energy conversion and energy storage in order to achieve greener and energy-wise wastewater processes. The application of the feasibility approach aims to reduce the operational costs of WWTPs with quality improvements in the treatment of

effluent, while the purpose of the analysis is to improve plant performances by introducing tertiary treatments [29].

In addition, scientific public warnings on the potential health problems and degradation of aquatic ecosystems due to the presence of traces of certain pollutants in treated wastewater, the so-called contaminants of emerging concern (CECs), mean that the focus of future research should be on newly combined methods for wastewater treatment in WWTPs [30,31] while research is already being done on the feasibility of upgrading WWTPs to remove CECs [32,33]. That is why international experts have recently been making significant efforts to research advanced treatment methods to remove CECs from wastewater [34]. Recent research further highlights the problem of the presence of antibiotics in wastewater and offers alternative techniques for their removal from WWTPs, but on an environmentally friendly basis [35]. The presence of an increased number of antibiotics in wastewater leads to microbial pathogens becoming resistant to antibiotics, which opens up a health and environmental problem of global proportions [36] and further actualizes the improvement of next-generation treatment processes.

In addition to the need to increase efficiency in the use of water resources, the issue of treated wastewater as an unconventional resource has been updated [37,38]. Apart from the traditional use of reclaimed wastewater for irrigation, advances in WWTP technology have made it possible to use it as drinking water [39]. Regarding increased supply and overcoming market oscillations and shortages [40,41], the environmental benefits of water reuse are obvious and indisputable [42].

The ecological perspective of society and the necessity for sustainable economic development have encouraged researchers to value natural resources, which was not the case until recently. Thus, in the research related to wastewater management, the economic assessment and evaluation of this resource have been updated, with the aim of connecting a natural resource (such as water) with the population and wider community's benefits [43], by determining its economic value. In these analyses, in the economic course of the WWTP project, the reduction of polluted water is calculated as a net effect, i.e., the ecological savings in pollution costs. In doing so, the general rule of achieving maximum economic efficiency and the condition that marginal costs are equal to marginal benefits is applied [44].

In order to determine financial feasibility, it is not sufficient to assess the project's justification for significant environmental impacts. Making a final decision on the implementation of such projects requires socio-economic evaluation, with a focus on environmental benefits. Sustainable wastewater management projects have an undeniable impact on the quality of the environment. In this regard, the main goal of our paper is to assess the financial and economic feasibility of the WWTP project in Nov Dojran, North Macedonia. This project's main aim was to determine the optimal technical solution for WWTP in Dojran municipality, and to improve municipal infrastructure, with significant environmental benefits as a result of improvements in treatment. The effects range from local, through regional, to global.

This paper is the result of research on a feasibility study for the improvement of wastewater treatment systems in Nov Dojran, Dojran municipality, North Macedonia, as a part of the project "Building Municipal Capacity for Project Implementation", funded by SIDA (Swedish International Development Cooperation Agency). It is organized into several sections. Following the introduction, in which we have articulated some aspects of the problem of water resource pollution and sustainable wastewater management, the second section provides a brief overview of the relevant literature. The literature review is focused on the issue of economic evaluation of WWTP's environmental benefits. In the third section, we present basic information regarding the current status in wastewater treatment in Nov Dojran, Dojran municipality in North Macedonia. The fourth section contains the methodology for conducting a financial and economic feasibility analysis of a WWTP project in Nov Dojran, as well as the inputs used for their implementation. In this section, we first compared and determined the most financially profitable variant of reconstruction and/or construction of WWTP in the municipality of Dojran. Three technical variants of WWTP (V1, V2, and V3) are analyzed, which we have previously identified. The

aim of the economic analysis was to determine whether Variant 3 (as the most financially advantageous option) has a positive environmental impact on society and whether it should be implemented. In this paper, with reference to some earlier research [45], environmental benefits quantification was performed using "shadow prices" for undesirable outputs—pollutants in wastewater, which are removed by treatment in WWTPs. The fifth section presents the basic results of the financial and economic feasibility analysis of the WWTP project. In the last section, we offer concluding remarks on this research and its limitations, as well as the possibilities for further research.

## 2. Literature Review

As environmental projects affect not only their investors but also the wider community, intentions to improve economic CBA (cost-benefit analysis) have recently been intensified by explicitly including environmental costs and benefits from the implementation of such projects in the analysis. Researchers have long pointed to the restrictiveness of the assessment, the narrowness and simplicity of assumptions in the original version of the CBA [46], and that the lack of economic methods for quantifying external influences (which the market does not take into account) causes the problem of incomplete evaluation of such projects [46,47]. Improved CBA analysis requires a broader analytical framework, in order to perceive each project individually from the perspective of potentially measurable and significant effects on the environment [48]. As is well known, the investment costs of such projects are known, so the key thing is to consider and evaluate a wide range of benefits (which exceed the character of financial ones), which is a challenge in an economic CBA. Thus, in economic terms, impacts on health, disease, and mortality can be valued [49,50], as "a cost avoided is a benefit". When a specific project brings benefits to those who are directly affected by a particular problem, as well as those who are not, then we are referring to the economic evaluation of environmental externalities and impacts on the ecosystem and their inclusion in the analysis [51]. However, it is quite clear that it is insufficient to use the income that determines the market as a measure of the overall social effects, but a social perspective of value determination of these effects is necessary [52].

According to the European Urban Waste Water Treatment Directive [53,54], all wastewater must be treated before being disposed of in nature, with the expectation that adequate wastewater treatment will improve its availability and consequently improve the environment [55]. In order to determine the economic feasibility of wastewater treatment, it is not sufficient to compare wastewater treatment costs with the costs of used water. A comprehensive economic analysis should compare the costs and benefits not only of water as an economic factor but also of water as an environmental public good. Therefore, one of the conditions for social, economic, and environmental sustainability is the implementation of a comprehensive CBA of adequate wastewater treatment projects. Moreover, the improved CBA, as a widely accepted economic instrument, should be a means of supporting the decision-making process in wastewater treatment projects.

As treated wastewater has become an important unconventional resource, some studies, using the concept of "shadow price", have contributed to the development of a methodology that can be used to assess internal (which is easy to monetize) but also external economic impacts [56,57].

This concept is used in some studies to quantify the environmental benefits of wastewater treatment [58,59], which reflect the true values of factors and products in socio-economic analyses and may differ from market values. In some studies, "shadow price" (implied costs of non-removal) have been calculated for basic wastewater pollutants [56], such as nitrogen (N), phosphorus (P), suspended particles (SP), biological oxygen demand (BOD), and chemical oxygen demand (COD). Some recent research uses this approach for different CBA inputs and outputs to determine different "weights", given that they have different environmental impacts [60]. Thus, environmental dimensions are integrated into the traditional techno-economic evaluation of the WWTP's efficiency; pollutants removed from wastewater are introduced into the analysis, which represent a smaller/larger individ-

ual impact on the environment [60]. Additionally, in order to monetize the benefits of wastewater treatment projects, such as environmental and health benefits, other methods of assessing these benefits are being developed [61] to overcome CBA constraints and include non-market benefits that are not easy to assess.

Precisely, the evaluation of external economic influences, often ensures that such projects are in the zone of economic justification. The methodology that is often applied to water resources for quantification of externalities (positive or negative) to the environment is CVM—the contingent valuation method [62–66]. Furthermore, in some studies, CBA is combined with methods of valuing non-market benefits, using the WTP (willingness to pay) or WTA (willingness-to-accept) approach, in order to determine the justification for public funding for WWTPs [67,68].

Additionally, some research has been done in order to make a scientific contribution to a comprehensive, expanded CBA in wastewater reuse [69,70]. Thus, guided by the importance of non-market benefits from projects of this type, some of the studies assessed the environmental benefits of using treated wastewater for various purposes [71,72]. These results can be used more widely—in optimizing the WWTPs capacity and the exploitation of treated wastewater in agriculture as an alternative to groundwater [73,74]. Assessment of the economic and social costs and benefits in the CBA is also valuable in the process of deciding on the WWTP's size because certain research shows that different sizes of these plants have different economic and social justification.

Economic CBA analysis has its application in research that from the perspective of costs and benefits considers possibilities of more rigorous standards of treated wastewater [75], by accounting for higher standard implementation costs with the benefits, while identifying the most economically efficient standard. The findings of these studies, based on a more comprehensive CBA, may have direct implications on wastewater management policy, as well as policies that are complementary to it or those with which there is a synergistic relationship.

On the other hand, it should be noted that some research indicates the possibility of controversial CBA results due to the monetization of environmental impacts, as in the case with the reduction of water eutrophication, due to the reduction of nitrogen and phosphorus [76].

## 3. Current Status in Wastewater Treatment: Nov Dojran, North Macedonia

Wastewater treatment system improvement in Nov Dojran, Dojran municipality, North Macedonia, aims to contribute to the process of implementation of European environmental standards, which defines the maximum permissible concentrations of pollutants in wastewater discharged into recipients. The existing system is not able to meet the standards regarding the standardized structure of treated water, as well as the required capacity.

Up to date, three settlements in the municipality of Dojran (Star Dojran, Nov Dojran, and Sretenovo) are included in the wastewater disposal system. In the municipality of Dojran, the fecal sewage network is divided into a primary and secondary network. The secondary sewerage network is used for receiving wastewater from households and their drain to the main collector. The main collector system (8340 m long) was built in 1989 and stretches along Lake Dojran on the Macedonian side. Along the collector, there are 10 pumping stations in which submersible pumps are installed, with which sewage is brought to the Toplec WWTP. The existing wastewater treatment plant—Toplec, was built in 1988 and is located in the suburb of Nov Dojran. The process of wastewater disposal ends with a treatment plant, from where the treated water is discharged into Dojran Lake. The plan is designed for 8000 equivalent residents and consists of two blocks, the first of which is technologically obsolete and out of use, while the second is in operation. For the second block, a reconstruction project was undertaken, in order to increase the efficiency of WWTP's work by replacing and supplementing the treatment technology. However, even after the reconstruction of the plant, the problem of sludge treatment had not been solved. Furthermore, the drainage network of atmospheric waters covers only a small

part of the Dojran municipality, and these atmospheric waters burden the fecal sewage and additionally complicate the problem. Thus, the increase of anthropogenic pressure in the area of the lake basin and the deterioration of the ecological condition of Dojran Lake intensify the need to take measures in which the principles of ecological sustainability are incorporated. Reconstruction of the existing plant and construction of a new wastewater treatment plant should improve the quality of surface and groundwater, as well as the quality of the soil of the wider region. Certainly, this investment should have a positive impact on the environment because it solves the long-standing problem of loading Dojran Lake with organic matter originating from municipal wastewaters.

To that end, it was necessary to analyze the condition of the existing treatment plant and assess the possibility of upgrading its current capacity, as well as analyze wastewater management solutions which would exceed the capacity of the (rehabilitated) existing treatment plant, through the construction of a new treatment plant. Certainly, the planned wastewater treatment plant for the municipality of Dojran envisages the treatment of municipal water, whereby quality should be ensured in accordance with the standards given in the EU Directive on Municipal Wastewater.

The selection of technical variants of wastewater treatment in the municipality of Dojran was made on the basis of several criteria, such as:

- Analysis of the system in the tourist season and beyond;
- Assessment of the condition and effectiveness of the existing WWTP and its combination with the new plant;
- Efficiency of treatment using different types of wastewater treatment technology;
- Size of the land area required to accommodate the treatment plant;
- Financial parameters for the proposed system (initial investment and funds required for plant maintenance).

Regarding the need to build/reconstruct WWTP in the municipality of Dojran, North Macedonia, three alternative technical solutions were considered to improve the wastewater treatment process for this municipality, namely:

- Variant A: a combination of the existing WWTP and a new moving bed biofilm reactor (MBBR) for wastewater treatment for 2000 equivalent inhabitants, which would be located next to the existing one, on a land area of 660 m$^2$;
- Variant B: a combination of the existing WWTP and the new sequencing batch reactor (SBR) for wastewater treatment for 2000 equivalent inhabitants, which would be located next to the existing one, on a land area of 900 m$^2$;
- Variant C: construction of a new MBBR for wastewater treatment for 6000 equivalent inhabitants, which would be located next to the existing plant, on a land area of 2400 m$^2$. In this particular case, it is a container modular plant, two two-stage MBBR-BNB bioreactors with a movable bearing, which includes an automatic mixer, a fine grate, and a sludge pump. It is planned that both reactors will be active in the tourist season, and only one outside the tourist season. The advantages of MBBR technology are multiple, as follows: a longer retention time of activated sludge (good for nitrification), the process does not require a secondary precipitator, reduction of sediment, does not take up a large land area, high flexibility in the operation of 30–70%, a two-stage biological process (medium and high load), increases in efficiency and adaptability to changing water flow, etc.

Estimated investment and operating costs, for all three presented variants are given in Table 1, as follows.

In order to select the optimal wastewater treatment technology in a particular case, a complex analysis of the proposed technical variants was performed, using the analytical hierarchy process (the AHP method), which is based on a mathematical and human approach, performing decomposition by hierarchy and enabling evaluation according to different criteria. This method considers several factors, such as initial investment,

operating costs, complexity facilities, and equipment, as well as the need for a professional management workforce. The AHP method was implemented in the following phases:

- Setting a target function (selection of a variant solution for wastewater treatment);
- Defining decision-making criteria (initial investment, operating costs, complexity of facilities and equipment, and the need for a professional management workforce);
- Selection of alternatives that achieve the target function (variants A, B, and C).

**Table 1.** Estimated investment and operating costs for variants A, B, and C.

| VARIANT A | Amount | VARIANT B | Amount | VARIANT C | Amount |
|---|---|---|---|---|---|
| 1. Investment in new MBBR (for 2000 equivalent inhabitants) | | 1. Investment in new SBR (for 2000 equivalent inhabitants) | | 1. Investment in new MBBR (for 6000 equivalent inhabitants) | |
| 1.1. Investment costs | 1,450,500.00 | 1.1. Investment costs | 1,456,000,00 | 1.1. Investment costs | 3,050,000.00 |
| 1.2. Operating costs (€/y) | 64,750.00 | 1.2. Operating costs (€/y) | 69,100.00 | 1.2. Operating costs (€/y) | 88,150.00 |
| 2. Existing WWPT | | 2. Existing WWPT | | | |
| 2.1. Reconstruction of existing facility | 2,000,000.00 | 2.1. Reconstruction of existing facility | 2,000,000.00 | | |
| 2.2. Operating costs | 101,500.00 | 2.2. Operating costs | 101,500.00 | | |

According to the results of the implemented AHP method, the following ranking of variant solutions was carried out:

- Variant A: 17.58%
- Variant B: 12.50%
- Variant C: 69.92%.

Following the analysis, it was determined that the V3 variant is technically the most favorable; this variant was the subject of financial and economic feasibility analysis in the remainder of our paper. As previously stated, the V3 variant involves the construction of a new WWTP with MBBR methodology, and the capacity of the plant is designed according to the estimated number of inhabitants of Dojran. Advantages of the MBBR wastewater treatment technology include:

- Longer retention time of activated sludge, which is good for nitrification;
- The process can take place without a secondary settler;
- Reduced settler production;
- The plant occupies a small land area;
- The capacity/space ratio occupied by the plant is maximized;
- High flexibility in the operation of the plant, 30–70% of the share are girders in relation to the volume of the tank;
- Two-stage biological process (high and medium load) increases efficiency and adaptability to changing wastewater inflows;
- The carrier material cannot be damaged—there are plants that are up to 20 years old, and still use the same girders;
- The thickness of the biofilm is controlled and maintained by continuous separation resulting from aeration and mixing; etc.

The chosen optimal solution for the wastewater treatment system in Nov Dojran, Dojran Municipality, consists of a central plant with MBBR-BNB reactors, in which wastewater is supplied by transferring from the separation shaft to the facility, primarily to the rough grate for storage of large waste. Following this phase, the water supply and purification are compensated in the equalization pool. The reactor itself constructively adapts to the process conditions, and also contains a charge inhabited by active biomass. After processing with an automatic mixer, passing through a fine grate, and carrying out the process of removing excess sludge from treated water (separator blades, aerobic digester, drainage, and drying),

filtered water is fed into a small tank where the microfiltration system creates "effluent" (filtered clean water). Effluent is then taken to the inspection chamber, from where it continues to the recipient. Figure 1 presents the Process Flow Diagram of Variant V3.

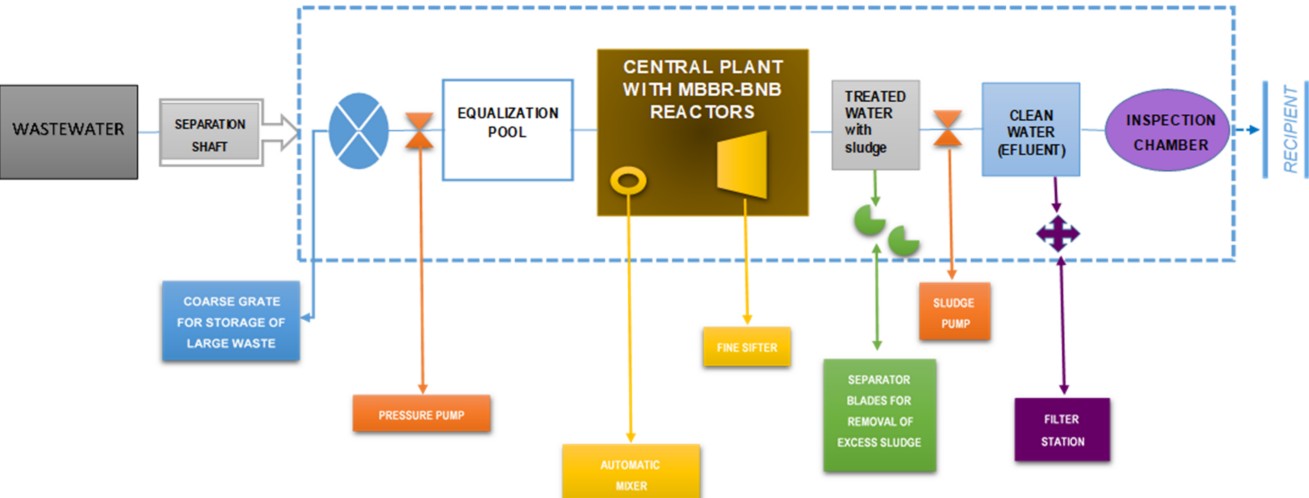

**Figure 1.** Process Flow Diagram of Variant V3.

Due to the high level of purification, effluent from the plant can be discharged into natural watercourses as it meets all quality standards. The efficiency of MBBR, depending on the water load of pollutants, is shown in Table 2.

**Table 2.** Efficiency of MBBR depending on water load with pollutants.

| Efficiency of Treatment | Load (gBOD5/m$^2$ day) |
| --- | --- |
| 75–80% | 20.0 |
| 80–85% | 10.0 |
| 85–80% | 6.0 |
| 90–95% | 4.5 |
| 95–100% | 2.5 |

## 4. Materials and Methods

In the part that follows, we present the methodological basis of the conducted financial and economic feasibility analysis of the technical variant V3 of the WWTP construction project in Nov Dojran, as well as inputs used for its implementation, procedure, and results of the conducted financial and economic analysis.

The financial and economic feasibility analysis of the wastewater treatment plant construction conducted in this paper was conducted in accordance with European methodology [77], with supporting documentation, as well as the authors' previous experiences in preparation of similar analyzes.

The financial and economic analysis is divided into several parts. Within the first part of the analysis, we performed a comparative financial analysis of the project's technical variants (V1, V2, and V3) for construction of the wastewater treatment plant (in Nov Dojran, North Macedonia), based on which we determined the most financially favorable variant for the investor. The financial analysis was conducted considering the following basic assumptions:

- Analysis was performed in euros;
- Analysis was conducted using real/constant prices;
- Starting year of the analysis is 2022;
- Construction period is 1 year;
- Observed period of project exploitation is 24 years (2023–2046);

- Final year of analysis is 2046;
- Discount rate is set at 4% [77].

In the second part of the analysis, we prepared a socio-economic analysis of the project, i.e., the justification for the project from the perspective of the wider social community was considered. Appropriate collection, treatment, and safe disposal of wastewater lead to significant benefits for the environment and human health, which are reflected in savings/reduction of non-response costs, or savings/reduction of costs that would certainly occur if certain measures, or, in this case, a concrete project for the construction of a wastewater treatment plant was not realized. The costs of non-response, in the case of a specific project, can be grouped into three:

- Adverse effects on human health are associated with reduced quality of drinking water and bathing/recreational water. They manifest in an increased number of diseases due to the reduced quality of drinking water and bathing water, an increased number of diseases due to unsafe food (contaminated fish, fruit, vegetables, and other agricultural products), increased risk of contracting a disease at work or during recreation in irrigated wastewater areas, increased health care burden, etc.;
- Negative impacts on the environment due to water and ecosystem degradation are manifested through reduced biodiversity, degraded ecosystems, bad odors, and increased GHG emissions, etc.;
- Possible negative effects on economic activities refer to a decrease in industrial productivity, agricultural productivity, the market value of crops, number of tourists or willingness to pay for tourist services, and reduced fish and shellfish catches or a reduction in their market value, etc.

In order to determine the socio-economic justification for the implementation of a specific project, we prepared a cost-benefit analysis, which projected and quantified the envisaged reductions in water pollution. We "faced" them with the estimated project costs and discounted to the time point of the beginning of the project. In this way, the calculation of certain economic indicators for project justification was completed, namely: economic net present value (ENPV), economic internal rate of return (EIRR), and economic benefit-cost ratio (EBCR).

Net present value (NPV) is an indicator that takes into account time preferences and represents the sum of net effects in the economic life of the project, reduced by discounting to the present moment, i.e., at the beginning of the investment.

Internal rate of return (IRR) is the rate at which the NPV of a project equals 0. The rate reflects the efficiency of the project, and the eligibility criterion is that it should be higher than the discount rate.

The benefit-cost ratio (BCR) shows how much net benefit can be achieved per unit of cost. It is calculated as the ratio of the discounted sum of all future benefits and the discounted sum of all costs.

After determining the above indicators, a conclusion is made on project justification from the socio-economic aspects and the need for project implementation. The basic methodological determinants for conducting the economic analysis for project justification were the following:

- The cost-benefit analysis was performed in such a way that the basic principles and rules on which the analysis is based are set in accordance with the principles and rules of the EC and international financial institutions;
- The transformation of the market into accounting (economic) prices was done with the help of a standard conversion factor;
- In order to reduce costs and benefits to the same base year, a discounting process is applied (according to the EU methodology for countries acceding to the EU, in CBA recommended discount rate is 5%);
- By applying the cost-benefit analysis, the above-mentioned indicators (EIRR, ENPV, and EBCR) of project evaluation from the socio-economic aspect are determined. By comparing the value of EIRR with OCC (opportunity cost of capital) and ENPV with 0,

we prepared an evaluation of the project for wastewater treatment plant construction from the socio-economic aspects.

The third part of the analysis contains a project sensitivity analysis. Namely, the previous indicators (EIRR, ENPV, and EBCR) were subjected to a sensitivity test, given the possible deviations of economic construction costs and economic benefits of construction due to changes in some of the key input parameters. The analysis ends with a review of the basic conclusions.

For better monitoring and understanding of the financial and economic evaluation process of the wastewater treatment plant construction, the analysis process is shown in Figure 2.

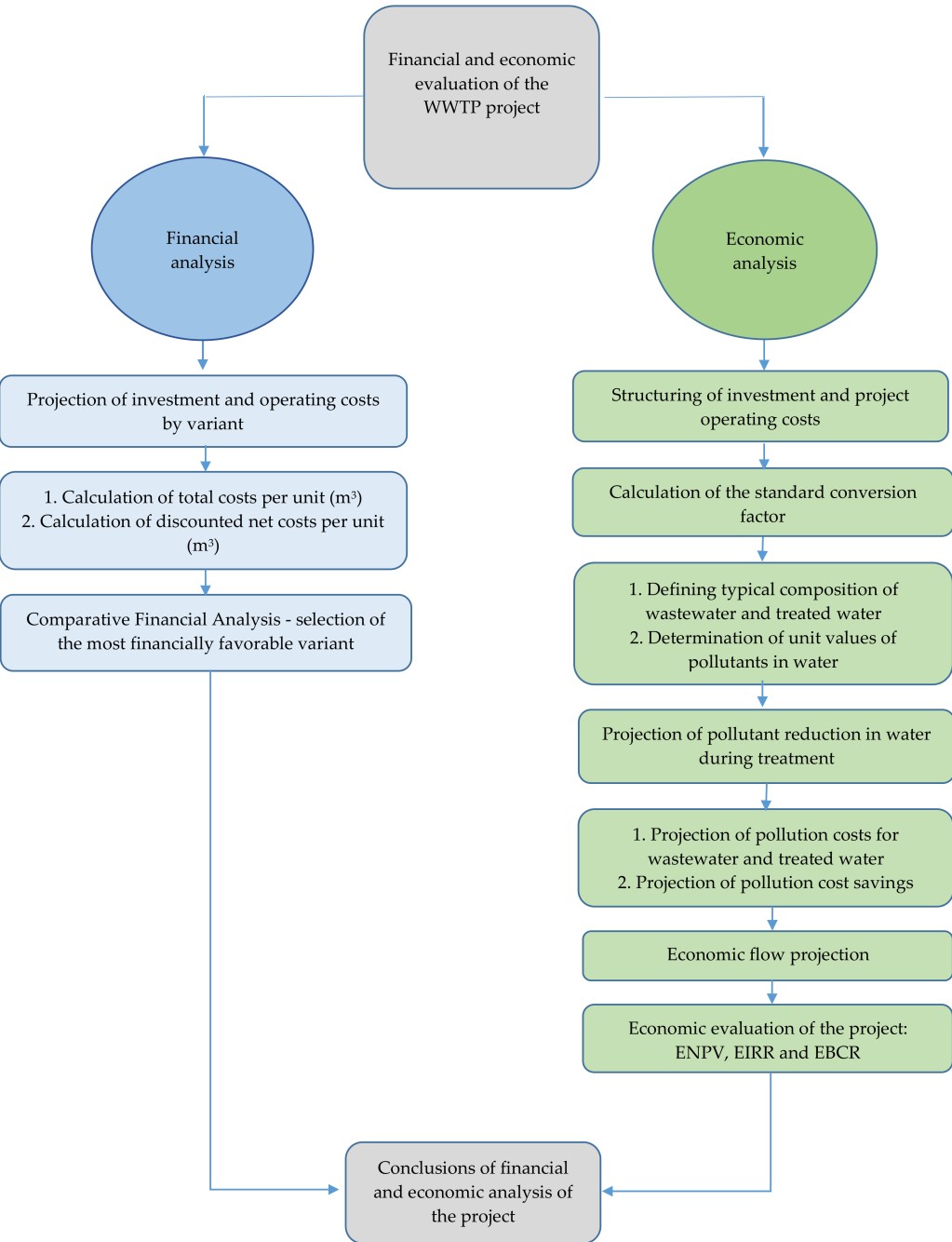

**Figure 2.** Methodological flowchart.

## 5. Results and Discussion

The first part of this section presents the quantitative procedure of the financial feasibility analysis of the planned wastewater treatment plant for the municipality of Dojran. Following the analysis procedure, we present the final results of the analysis with discussion. In the second part of this section, the procedure for quantifying the expected economic benefits from this project is presented within the economic feasibility analysis of the project, while the final results of the analysis with discussion are presented at the end of this part. In the third and final part of this section, we use an economic flow projection to conduct a project sensitivity analysis, with results concluding this part of the section.

### 5.1. Financial Analysis

As already mentioned, the aim of the financial analysis was to compare and determine the most financially viable variant of reconstruction and/or construction of a wastewater treatment plant in Nov Dojran, Dojran municipality. In the financial analysis, three technical variants (V1, V2, and V3) were analyzed, which were previously determined. Financial feasibility analysis in this case is a comparative analysis of investment costs and operating costs of individual technical variants. Within this analysis, we prepared a projection of investment and operating costs of the project by variants, and then their discounting (based on the defined discount rate), in order to reduce the cost categories to a common value.

Investment costs and operating costs are determined in advance for each of the defined technical variants. Table 3 gives projections of these costs by variants and calculated total costs (TC), total discounted net costs (DNC), and the amount of these costs (total and discounted) per measured unit (in $m^3$). The observed period of project exploitation is 24 years. In order to rationalize the space, all tables in the paper have been reduced by not showing a certain number of years.

**Table 3.** Projection of investment and operating costs of the WWTP project.

| Year | Variant V1 | | | Variant V2 | | | Variant V3 | | |
|---|---|---|---|---|---|---|---|---|---|
| | Investment | Operating Costs | Total Costs | Investment | Operating Costs | Total Costs | Investment | Operating Costs | Total Costs |
| 1 | 3,450,500 | | 3,450,500 | 3,456,000 | | 3,456,000 | 3,050,000 | | 3,050,000 |
| 2 | | 166,250 | 166,250 | | 170,650 | 170,650 | | 88,150 | 88,150 |
| 3 | | 179,297 | 179,297 | | 184,043 | 184,043 | | 95,068 | 95,068 |
| 4 | | 185,573 | 185,573 | | 185,573 | 185,573 | | 185,573 | 185,573 |
| 5 | | 192,068 | 192,068 | | 192,068 | 192,068 | | 192,068 | 192,068 |
| 6 | | 198,790 | 198,790 | | 198,790 | 198,790 | | 198,790 | 198,790 |
| 7 | | 205,748 | 205,748 | | 205,748 | 205,748 | | 205,748 | 205,748 |
| . . . | | | | | | | | | |
| 23 | | 334,996 | 334,996 | | 334,996 | 334,996 | | 334,996 | 334,996 |
| 24 | | 345,046 | 345,046 | | 345,046 | 345,046 | | 345,046 | 345,046 |
| 25 | | 355,398 | 355,398 | | 355,398 | 355,398 | | 355,398 | 355,398 |
| | TC = | 9,609,689 | | TC = | 9,624,334 | | TC = | 9,046,859 | |
| | DNC = | 6,866,928 | | DNC = | 6,880,503 | | DNC = | 6,334,744 | |
| | TC/$m^3$ = | 1.54 | | TC/$m^3$ = | 1.54 | | TC/$m^3$ = | 1.45 | |
| | DNC/$m^3$ = | 1.10 | | DNC/$m^3$ = | 1.10 | | DNC/$m^3$ = | 1.01 | |

In this section, we present the final results of the financial and economic feasibility analysis of the WWTP project. Table 4 shows the results of the comparative financial analysis of the considered technical variants of the construction of WWTP.

**Table 4.** Recapitulation of comparative financial analysis for offered technical variants of WWTP.

| Indicators | Variant 1 | Variant 2 | Variant 3 |
|---|---|---|---|
| Total costs (TC)—in EUR | 9,609,689 | 9,624,334 | 9,046,859 |
| Discounted net costs (DNC)—in EUR | 6,866,928 | 6,880,503 | 6,334,744 |
| Total costs per unit measure (TC/m$^3$)—in EUR/m$^3$ | 1.54 | 1.54 | 1.45 |
| Discounted net costs per unit measure (DNC/m$^3$)—in EUR/m$^3$ | 1.10 | 1.10 | 1.01 |

From Table 4 it can be concluded that the most financially favorable variant for the implementation of this project is Variant 3. The difference between the first two variants (Variant 1 and Variant 2) is minimal, while Variant 3 has the lowest total and discounted costs, and therefore the lowest cost per cubic meter of treated water.

The effects of the wastewater treatment plant construction project—costs and benefits—in the period 2022–2046 are discounted at the rate (5%) and reduced to a common denominator, i.e., they are expressed in current monetary units, as shown in Table 5.

**Table 5.** Economic flow projection of the WWTP project.

| Year | Investment | Operating Costs | Ecological Savings | Residual Value | Net Effects |
|---|---|---|---|---|---|
| 1 | 2,775,500 | | | | −2,775,500 |
| 2 | | 80,217 | 457,368 | | 377,151 |
| 3 | | 86,512 | 475,073 | | 388,561 |
| 4 | | 89,540 | 492,512 | | 402,972 |
| 5 | | 92,674 | 510,590 | | 417,917 |
| 6 | | 95,917 | 529,333 | | 433,415 |
| 7 | | 99,274 | 548,763 | | 449,489 |
| . . . | | | | | |
| 23 | | 161,638 | 917,366 | | 755,728 |
| 24 | | 166,487 | 946,445 | | 779,959 |
| 25 | | 171,481 | 976,447 | 1,040,813 | 1,845,778 |

The issue of the price of communal services, which represents the lower point of profitability or the threshold of this project's profitability, includes, in addition to the economic component, various political, social, environmental, and other influences. At this level of our analysis, it can be concluded that the price that ensures the lower point of profitability, covering the total costs for Variant 3 (investment/CAPEX + operational/OPEX), is 1.45 EUR/m$^3$; if their discounted value is considered, then this amount is 1.01 EUR/m$^3$. The price that ensures the lower point of profitability, i.e., a sufficient amount to cover only the operating costs of the plant (which is one of the possibilities), is 0.96 EUR/m$^3$; if their discounted value is considered, then this amount is 0.57 EUR/m$^3$.

We emphasize that the motives for investing in communal infrastructure are not usually exclusively based on the fact that the investment is covered by an increase in prices of services paid by end users, who can bear more or less of the total costs. Once the WWTP construction in the municipality of Dorjan is complete, the final decision is up to the relevant authorities (within the municipality and/or utility company). The decision on possible corrections to existing prices, i.e., an increase in fees for wastewater treatment, should be made after a comprehensive analysis of the situation, which would consider many other factors (current prices, social status, wider economic context, experiences in the region, etc.).

Considering these factors in their analysis of the financial viability of municipal and domestic wastewater treatment plants, Fitriani et al. pointed out that the justification for these plants depends on the ability to pay (ATP) and willingness to pay (WTP) of users, and the amount of the discount rate selected in the analysis. Therefore, they conclude that subsidies from local governments can be helpful in the construction and operation of these plants in the initial phase, to ensure financial justification [78]. In this regard,

recent research recommends the minimum flow guarantee (MFG) incentive, as a suitable risk mitigation strategy, in order to enable proper distribution of risks and benefits (for the public and private sector), and at the same time achieve socio-economic benefits and financial profitability [79]. A similar recommendation was made after the CBA analysis of decentralized wastewater reuse systems, which showed that the economic, environmental, and social benefits of these systems are indisputable, but their financial justification is questionable, due to the low rate charged for reclaimed water [80].

### 5.2. Economic Analysis

The economic cost-benefit analysis involves examining the impact of a wastewater treatment plant construction project on the economic well-being of the wider community. The subject of economic analysis in our paper is Variant 3 which, according to the conducted financial analysis, turned out to be the most financially favorable variant. As previously stated, the purpose of economic analysis is to prove that this project has a positive contribution to society as a whole and that it should therefore be implemented.

The economic benefits of the project should be greater than the project costs, which is reflected in the positive economic net present value, cost-benefit ratio (greater than 1), and economic internal rate of return, which should be higher than the discount rate (used to calculate the net present value).

### 5.2.1. Expected Project Costs

Project costs, considered within the economic analysis, are defined as investment costs and operating costs of maintenance and management of the new facility. Investment costs and operating costs have already been determined, and their structure is presented in Table 6.

**Table 6.** Structure of investment costs and annual operating costs of the project.

| Type of Work | Amount (in EUR) | Type of Cost | Amount (in EUR) |
|---|---|---|---|
| WWTP construction | 2,400,000 | Electricity costs | 29,800 |
| Press construction | 300,000 | Earnings costs | 10,000 |
| Dryer construction | 350,000 | Dryer maintenance costs | 47,110 |
| | | Press maintenance costs | 1240 |
| Total | 3,050,000 | Total | 88,150 |

It should be considered that operating costs increased in the observed period (2023–2046), in accordance with estimated growth in the gross domestic product (GDP) in North Macedonia, as follows in Table 7.

**Table 7.** GDP growth rate forecast.

| 2022 [1] | 2023 [2] | 2024–2030 [3] | 2031–2046 [4] |
|---|---|---|---|
| 4% | 3.7% | 3.5% | 3% |

[1] [81] European Bank for Reconstruction and Development, https://www.ebrd.com/where-we-are/north-macedonia/overview.html (accessed on 1 November 2021). [2] Ibid. [3] [82] Statista, https://www.statista.com/map/europe/north-macedonia/ (accessed on 2 November 2021). [4] Authors estimation.

When conducting economic analysis, economic prices of investments and costs should be used, and the conversion of financial into economic prices is usually done using sectoral conversion factors, if any. When sector-specific conversion factors are not available, the standard conversion factor (SCF) is applied based on average differences between domestic and international prices due to trade tariffs and barriers, which can be estimated based on foreign trade statistics, using the following equation:

$$SCF = (M + X)/(M + X + Tm) \tag{1}$$

where: M—total import, X—total export, Tm—total value of duties on import.

Based on the available data, we calculate the standard conversion factor (SCF), as follows in Table 8.

**Table 8.** Standard conversion factor calculation.

| Description | Amount (Million $) |
|---|---|
| Total import | 9446 [1] |
| Total export | 7198 [2] |
| Total value of duties on import | 1631 [3] |
| Standard conversion factor (SCF) | 0.91 |

[1] [83] State Statistical Office, Republic of North Macedonia, Data for 2019, https://www.stat.gov.mk/ (accessed on 3 November 2021). [2] Ibid. [3] [84] Customs Administration, Republic of North Macedonia, https://customs.gov.mk/index.php/mk/ (accessed on 13 October 2021).

5.2.2. Expected Project Benefits

Environmental pollution in Europe and beyond is increasing, causing huge risks to human health and life. These adverse effects produce heavy costs for health services, large negative effects due to prematurely lost lives (based on the principle of the value of statistic life—VSL), and damage to the entire environment and the economy. Therefore, investments in the reduction of environmental pollution result in savings in these costs, and thus the economic savings of these investments can be quantified.

Monetization of environmental benefits was calculated using defined "shadow prices" for undesirable outputs, i.e., pollutants in wastewater. These prices reflect the assessment of the benefits/avoided damage to the environment after wastewater treatment, i.e., the value of the ecological damage if the treatment was not performed.

Given that investments in the construction of wastewater treatment plants directly lead to a reduction in emissions of certain types of pollutants into the environment, the most acceptable method for economic analysis of these investments is to determine the "damage" caused by certain pollutants per unit of emission (usually in kilograms). This approach to economic analysis is not an easy task, as it is necessary to establish a clear correlation between the emission of individual pollutants in water and the harmful effects on health, the environment, and the economy, which they cause.

In North Macedonia, but also in most more developed EU countries, no individual studies of these values have been conducted, but there are some relevant studies and methodologies that have adequately investigated this issue, determining the values of pollutant emissions in water, by determining their "shadow prices". For the purposes of economic analysis, this paper uses recommendations from relevant sources [85] in which the unit values of relevant pollutants ("shadow prices") were determined, based on one of the previous studies [56], as presented in Table 9.

**Table 9.** "Shadow prices" of relevant pollutants in water [1].

| | Nitrogen | Phosphorus | Suspended Particles | BOD [2] | COD [3] |
|---|---|---|---|---|---|
| "Shadow prices" (EUR/kg) | 35.2 | 82.5 | 0.01 | 0.03 | 0.21 |

[1] Defined unit values per kilogram of pollutants emitted in water, i.e., "shadow prices" (Table 9) represent the benefits that are achieved by reducing the pollutants in water. These benefits represent avoided costs due to reduced environmental degradation, hospital treatment costs, cost of lost lives, etc. Defining these unit costs is the subject of special expert research and analysis. The authors of the paper have examined numerous studies that have dealt with, among other things, the determination of these values, i.e., the impact of pollutants from wastewater on the environment and human health. Similar results were presented in the review papers, and the authors opted for unit values from the research, which is also referred to in the paper. [2] Biological Oxygen Demand. [3] Chemical Oxygen Demand.

From Table 9, it is clear that the environmental benefits associated with the removal of nitrogen and phosphorus from wastewater are by far the greatest, while the benefits, i.e., the harmful effects of other pollutants, are significantly smaller. Therefore, there are significant

differences in the "shadow prices" of relevant pollutants in water. In order to successfully conduct the economic analysis and quantify the effects of the wastewater treatment plant project, it is necessary to perform pollution projections with relevant pollutants, for the case of "no investment" and "with investment". These projections were carried out considering the annual quantities of treated water in the plant and its level of pollution.

The amount of wastewater that would be treated in this plant was determined at the level of 18.75 $m^3$/h out of season, or 56.25 $m^3$/h in season (June, July, August). Based on that, the total annual amount of wastewater that would be treated, out of about 245 million liters, was calculated, with a trend of a slight increase, in accordance with the established projection of population growth in the municipality of Dojran.

Pollution levels of wastewater and treated water are defined on the basis of the typical composition of these waters, taken from the international literature [57,86,87], as well as the relevant regulations governing this area, which are shown in Table 10.

**Table 10.** Typical concentration values of relevant pollutants in wastewater and treated water.

| Water Composition | Nitrogen | Phosphorus | Suspended Particles | BOD | COD |
|---|---|---|---|---|---|
| Wastewater—concentration mg/L | 40 | 12 | 150 | 250 | 500 |
| Treated water—concentration mg/L [1] | 15 | 2 | 35 | 25 | 125 |

[1] Pollutant values are in line with the requirements of the European Union [53,54]. Council Directive Concerning Urban Wastewater Treatment (91/271/EC and 98/15 EC).

According to the previously mentioned data, projections of the total quantities of pollutants are calculated, considering the structure for both wastewater and treated water. After determining the total amount of pollutants for wastewater in treated water, projections for reducing the number of pollutants after treatment in the wastewater treatment plant were determined in the observed project operation period of 24 years (2023–2046), as follows in Table 11. The amount of reduction in pollutants after treatment in WWTP increases slightly over time because the amount of treated water also increases in accordance with the projected increase in population in the area analyzed.

**Table 11.** Projection of reduction in pollutants after treatment in WWTP.

| Year | Nitrogen (kg) | Phosphorus (kg) | Suspended Particles (kg) | BOD (kg) | COD (kg) |
|---|---|---|---|---|---|
| 1 | 6131 | 2453 | 28,204 | 55,181 | 91,969 |
| 2 | 6141 | 2457 | 28,250 | 55,272 | 92,120 |
| 3 | 6151 | 2461 | 28,297 | 55,363 | 92,272 |
| 4 | 6162 | 2465 | 28,344 | 55,455 | 92,425 |
| 5 | 6172 | 2469 | 28,390 | 55,546 | 92,577 |
| 6 | 6182 | 2473 | 28,437 | 55,638 | 92,730 |
| 7 | 6192 | 2477 | 28,484 | 55,730 | 92,883 |
| . . . | | | | | |
| 23 | 6358 | 2543 | 29,245 | 57,219 | 95,365 |
| 24 | 6368 | 2547 | 29,293 | 57,313 | 95,522 |
| 25 | 6379 | 2551 | 29,342 | 57,408 | 95,680 |

The percentage structure of the reduction of pollutants in water after treatment in the WWTP is given in Figure 3. This clearly indicates that the largest reductions in the number of pollutants were provided by BOD and COD.

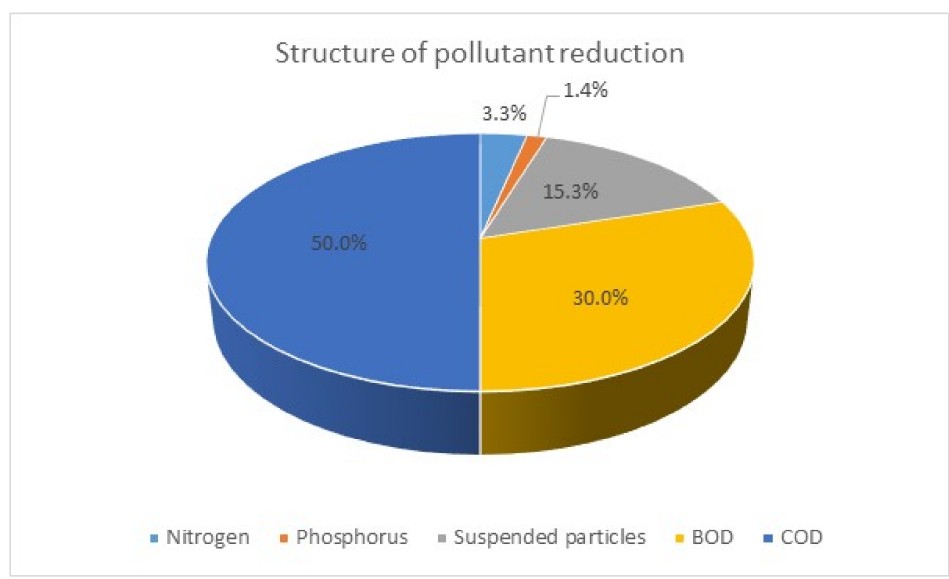

**Figure 3.** Structure of pollutant reduction after treatment in WWTP.

After determining the degree of pollution, i.e., the number of pollutants in wastewater and treated water, we performed an evaluation of pollution costs based on "shadow prices" for the period 2023–2046, as presented in Table 12.

**Table 12.** Projection of pollution costs for wastewater and treated water (in EUR).

| Year | Nitrogen | Phosphorus | Suspended Particles | BOD | COD | Total |
|---|---|---|---|---|---|---|
| **WASTEWATER** | | | | | | |
| 1 | 345,312 | 242,798 | 368 | 12,692 | 12,263 | 613,432 |
| 2 | 359,717 | 252,926 | 383 | 13,221 | 12,774 | 639,021 |
| 3 | 373,642 | 262,717 | 398 | 13,733 | 13,269 | 663,758 |
| 4 | 387,357 | 272,360 | 413 | 14,237 | 13,756 | 688,123 |
| 5 | 401,576 | 282,358 | 428 | 14,760 | 14,261 | 713,382 |
| 6 | 416,317 | 292,723 | 444 | 15,301 | 14,784 | 739,568 |
| 7 | 431,598 | 303,468 | 460 | 15,863 | 15,327 | 766,716 |
| . . . | | | | | | |
| 23 | 721,502 | 507,306 | 769 | 26,518 | 25,622 | 1,281,717 |
| 24 | 744,373 | 523,387 | 793 | 27,359 | 26,434 | 1,322,346 |
| 25 | 767,969 | 539,978 | 818 | 28,226 | 27,272 | 1,364,263 |
| **TREATED WATER** | | | | | | |
| 1 | 129,492 | 40,466 | 86 | 1269 | 3066 | 174,379 |
| 2 | 134,894 | 42,154 | 89 | 1322 | 3194 | 181,653 |
| 3 | 140,116 | 43,786 | 93 | 1373 | 3317 | 188,685 |
| 4 | 145,259 | 45,393 | 96 | 1424 | 3439 | 195,611 |
| 5 | 150,591 | 47,060 | 100 | 1476 | 3565 | 202,792 |
| 6 | 156,119 | 48,787 | 103 | 1530 | 3696 | 210,235 |
| 7 | 161,849 | 50,578 | 107 | 1586 | 3832 | 217,953 |
| . . . | | | | | | |
| 23 | 270,563 | 84,551 | 179 | 2652 | 6405 | 364,351 |
| 24 | 279,140 | 87,231 | 185 | 2736 | 6608 | 375,900 |
| 25 | 287,988 | 89,996 | 191 | 2823 | 6818 | 387,816 |

Table 13 shows the savings of pollution costs, as the difference between the costs of pollution for wastewater and the costs of pollution for treated water.

**Table 13.** Projection of savings in pollution costs (in EUR).

| Year | Nitrogen | Phosphorus | Suspended Particles | BOD | COD | Total |
|------|----------|------------|---------------------|-----|-----|-------|
| 1 | 215,820 | 202,331 | 282 | 11,423 | 9197 | 439,053 |
| 2 | 224,823 | 210,772 | 294 | 11,899 | 9581 | 457,368 |
| 3 | 233,526 | 218,931 | 305 | 12,360 | 9951 | 475,073 |
| 4 | 242,098 | 226,967 | 316 | 12,813 | 10,317 | 492,512 |
| 5 | 250,985 | 235,298 | 328 | 13,284 | 10,695 | 510,590 |
| 6 | 260,198 | 243,936 | 340 | 13,771 | 11,088 | 529,333 |
| 7 | 269,749 | 252,890 | 353 | 14,277 | 11,495 | 548,763 |
| . . . | | | | | | |
| 23 | 450,939 | 422,755 | 589 | 23,866 | 19,216 | 917,366 |
| 24 | 465,233 | 436,156 | 608 | 24,623 | 19,825 | 946,445 |
| 25 | 479,981 | 449,982 | 627 | 25,404 | 20,454 | 976,447 |

Figure 4 shows the environmental benefits of pollutant removal after treatment in WWTP. The highest environmental benefit of 49.2% in total environmental benefits is related to nitrogen as its estimated "shadow price" is relatively high (immediately after phosphorus). Phosphorus is the next polluter in the structure of environmental benefits with 46.1% whose estimated "shadow price" is the highest. Environmental benefits from the removal of BOD and COD (whose "shadow prices" are relatively low) are significantly lower—2.6% and 2.1%, respectively, despite the removal of significant amounts of these pollutants during treatment. The situation is similar with suspended particles, which have the lowest "shadow price", making a small contribution to environmental benefits, regardless of the fact that solid amounts of this pollutant have been removed.

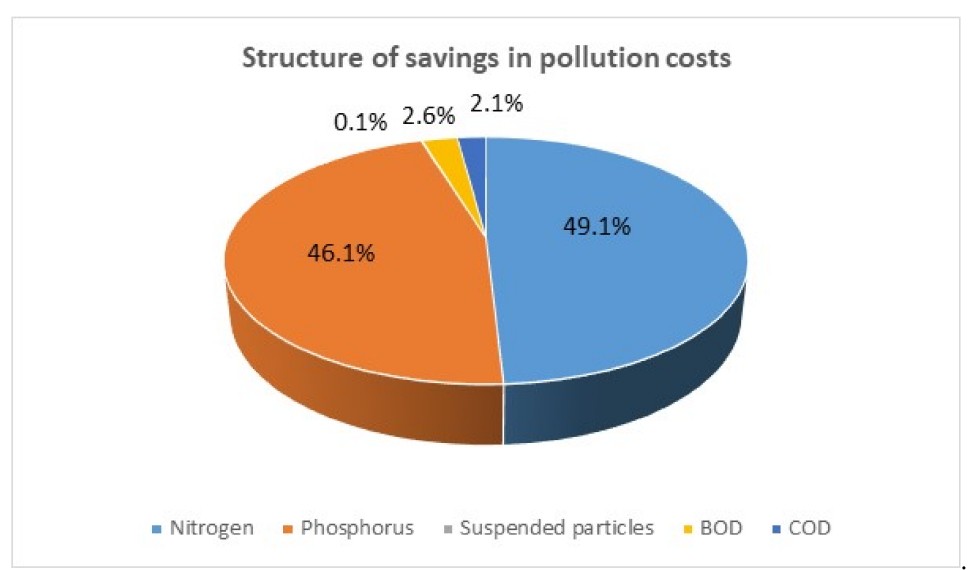

**Figure 4.** Structure of savings in pollution costs.

The relationship between investment costs and environmental effects/savings in pollution costs was addressed by Karczmarczyk et al. on the example of wastewater treatment from households in unsewered areas. Their recommendation is the use of on-site treatment systems, as opposed to centralized systems. In addition, the authors concluded that different technologies and on-site treatment systems provide different treatment efficiencies and consequently different environmental effects. In principle, the lower investment costs of these systems mean lower treatment efficiency, i.e., the absence of full environmental effects. In this regard, their future recommendation is a hybrid method (activated sludge supported with biological film) on-site wastewater treatment system, which achieves high environmental effects and the highest NPV [88].

Residual value of the project was estimated based on the following equation:

$$Y = (A/x) \times (x - v) \tag{2}$$

where: Y—residual value of project, A—investment value, x—physical lifetime of the project and v—analysis period.

According to the total investment value mentioned above and the estimated lifetime of the WWTP (40 years), the residual value is calculated (in the amount of EUR 1,040,813) and it is included in the economic flow of the project in the last observed year of the projection.

Economic analysis results, i.e., the basic indicators/indicators of economic justification for the investment are shown in Table 14.

**Table 14.** An overview of the economic feasibility indicators of the WWTP project.

| Economic Viability Indicators | Value |
| --- | --- |
| Economic Net Present Value (ENPV) | 4,612,349 EUR |
| Economic internal rate of return (EIRR) | 16.38% |
| Economic Benefit-Cost Ratio (EBCR) | 2.11 |

We conclude that investment in the wastewater treatment plant construction project has a satisfactory economic justification, as the EIRR is higher than the OCC (EIRR = 16.38%), the ENPV is greater than 0 (ENPV = 4,612,349 EUR), and the EBCR is greater than 1 (EBCR = 2.11). Therefore, our analysis showed that the direct benefits of this investment are savings in the cost of environmental pollution. From the analyzed socio-economic effects generated by this investment, it can unequivocally be concluded that this investment has full socio-economic justification.

Recent study results, based on an existing treated water reuse project, suggest that an assessment of the usual benefits of projects of this type (e.g., increased availability of water for agricultural irrigation) is usually not sufficient to justify these projects. Using CBA, Arena et al. concluded that in almost all project feasibility scenarios, in order for a project to be economically viable, environmental benefits must not only be included but also benefits that are not currently used [89].

### 5.3. Project Sensitivity Analysis

Considering that during the evaluation of the efficiency of the WWTP construction project, "future" values are used, which cause a certain greater or lesser degree of uncertainty of the obtained results, we performed a project sensitivity analysis, which determined the project profitability threshold by varying the following key parameters: investment costs and discount rates. In this regard, in the sensitivity analysis, sensitivity tests were performed with the following assumptions:

- Scope of work ±10% and ±20%;
- Investment costs +10% and +20%;
- Discount rate of 6%, 9% and 10%.

After the projections of economic flows within the sensitivity analysis, the recapitulation of the sensitivity analysis results is presented in Table 15, with the predefined assumptions considered during the performance of this project's sensitivity analysis.

**Table 15.** Sensitivity analysis results of the WWTP project.

| No. | Type of Test | EIRR Condition: EIRR > OCC | ENPV Condition: ENPV > 0 | EBCR Condition: EBCR > 1 |
|---|---|---|---|---|
| 1. | SCOPE OF WORK | | | |
| | Scenario 1: Base scenario | 16.38% | 4,612,349 | 2.11 |
| | Scenario 2: Scope of work decreased by 10% | 14.52% | 3,766,155 | 1.91 |
| | Scenario 3: Scope of work decreased by 20% | 12.59% | 2,919,962 | 1.70 |
| | Scenario 4: Scope of work increased by 10% | 18.20% | 5,458,542 | 2.31 |
| | Scenario 5: Scope of work increased by 20% | 19.98% | 6,304,735 | 2.52 |
| 2. | INVESTMENT COST | | | |
| | Scenario 1: Base scenario | 16.38% | 4,612,349 | 2.11 |
| | Scenario 2: Investment growth by 10% | 15.00% | 4,378,751 | 1.99 |
| | Scenario 3: Investment growth by 20% | 13.82% | 4,145,153 | 1.88 |
| 3. | DISCOUNT RATE | | | |
| | Scenario 1: Base scenario—discount rate 5% | 16.38% | 4,612,349 | 2.11 |
| | Scenario 2: Discount rate 6% | 16.38% | 3,794,860 | 1.96 |
| | Scenario 3: Discount rate 8% | 16.38% | 2,525,342 | 1.69 |
| | Scenario 4: Discount rate 10% | 16.38% | 1,610,343 | 1.47 |

Based on the results of the sensitivity analysis of the project presented in Table 15, it can be concluded that the sensitivity analysis proved that the project is resistant to all real changes in input parameters, i.e., all indicators remain in the cost-effectiveness zone, which further strengthens the belief in the justification and necessity of investments in the WWTP.

## 6. Conclusions

This paper presented the results of the CBA implemented with the example of the wastewater treatment plant project in the municipality of Dojran, North Macedonia. The financial CBA has shown that Variant 3 has the lowest total and discounted costs, and therefore the lowest cost per m$^3$ of treated water and is the most financially advantageous.

In the economic analysis of this investment, the ecological externalities of the project were monetized, starting from the assumption that the wastewater treatment process is a process that has the desired output—treated water—but also undesirable outputs, such as phosphorus, nitrogen, suspended particles, etc. Their estimated value, determined by "shadow prices", actually represents the avoided damage/costs, i.e., the realized benefit/income for the environment as a result of the removal of pollutants during treatment in the WWTP. The difference between the pollution costs for wastewater and the pollution costs for treated water represents the realized savings in the cost of pollution, i.e., environmental benefits. As the estimated "shadow price" for nitrogen is high (immediately after phosphorus), environmental benefits of about 49% are associated with this pollutant in wastewater. Immediately after nitrogen, the estimated environmental benefits of about 46% are bound to phosphorus, whose estimated "shadow price" is the highest. Although wastewater treatment removes significant amounts of BOD and COD, due to relatively low "shadow prices", the contribution of these pollutants to environmental benefits is low. The same can be concluded for suspended particles. The final results of the conducted economic cost-benefit analysis unequivocally indicate the full socio-economic justification for the investment. At the same time, the sensitivity analysis proved that the project is resistant to possible changes in input parameters over time.

We recognize that the research presented in our paper has a theoretical and practical contribution. Our paper has demonstrated the justification for using an improved original version of the CBA in WWPTs, by quantifying environmental benefits (which are not recognized by the market) in order to overcome the problem of incomplete evaluation of such projects. The paper presents the justification approach of WWPTs, based on the

assessment of environmental benefits, which should provide strong support to policy makers in the decision-making process, as well as the choice of sustainable technologies (such as MBBR technology) focusing on their high positive impact on the environment. This is the practical contribution of this paper. These projects improve municipal infrastructure, and in addition, environmental benefits are transmitted from the local, through the regional, to the global level.

Due to the frequent need for developing countries, in particular, to define socio-economically acceptable prices (usually lower than the market) for similar projects, it is recommended that the financial justification for the project be provided by budget funds and various incentives/subsidies from governments and local governments during construction and the operation of the plant itself. In addition, in such projects, socio-economic benefits have a higher specific weight compared to purely financial benefits, and financial analysis should not serve as an exclusive basis for the decision-making process on project implementation.

Certain limitations in the analysis may partly relate to the suitability of individual input data for this particular case (e.g., unit savings values, the typical structure of pollutants in water), although they are based on relevant international literature sources. This is a consequence of the lack of research of this kind in the Western Balkans region, whose results would be more adapted to the conditions of this project. In addition, a clear definition of the project's financing sources, as well as the determination of wastewater tariffs, would enable specification and concretization of the results of the financial analysis. However, we believe that these limitations cannot fundamentally question the results of our analysis. Rather, they could serve as a resource for future research.

**Author Contributions:** Conceptualization, J.Ć., M.K. and S.L.; methodology, J.Ć., M.K. and R.V.; validation, J.Ć., M.Ž. and R.V.; formal analysis, J.Ć., M.Ž. and A.Ž.; data curation, A.Ž. and J.C.; writing—original draft preparation, J.Ć., M.K., S.L. and M.Ž.; writing—review and editing, J.Ć., M.K., R.V. and A.Ž.; visualization, M.Ž. and J.C.; supervision, J.Ć., M.K. and S.L. All authors have read and agreed to the published version of the manuscript.

**Funding:** This paper is the result of research on the feasibility study for the improvement of WWTP in Nov Dojran, North Macedonia, as a part of the project "Building Municipal Capacity for Project Implementation", funded by SIDA. Implementing partner on the project is UNDP, North Macedonia, funding number RFQ 31/2021.

**Institutional Review Board Statement:** Not applicable.

**Informed Consent Statement:** Not applicable.

**Data Availability Statement:** They may be made available on request from the corresponding author.

**Conflicts of Interest:** The authors declare no conflict of interest.

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
