# Peer review of "Financial and Economic Investment Evaluation of Wastewater Treatment Plant"

_water, doi:10.3390/w14010122_

Round 1

Reviewer 1 Report

The article “Financial and economic investment evaluation in wastewater treatment plant” reports a cost-benefit analysis of different WWTP variants in North Macedonia, also considering (and monetizing) environmental aspects. The general topic of the article is novel and is in the scope of Water journal; however, its structure is not balanced throughout the different sections and, in addition, some relevant integrations are needed to enhance its value. I consequently recommend a major revision focusing on the following aspects:

  1. Introduction: the authors should mention energy consumption issues in WWTPs and the possibility to improve energy balances through anaerobic digestion and (more in general) integration of renewable energy (see 10.3390/en13184780 and 10.1016/j.enconman.2021.114214).
  2. Line 134: define acronyms at their first appearance in the text (CBA).
  3. Section 3, lines 209-224: these paragraphs are too general, please delete them.
  4. Section 3 is too general as well, please revise it to make it more quantitative; is the current WWTP applying a primary, secondary or tertiary treatment? Which technologies are used? Which is expected pollutant removal (COD, BOD, N, P, TSS) in the original and in the modified configuration? This is a key point to understand the concepts proposed in the overall manuscript.
  5. Section 4.2: a process diagram schematically representing the 3 analyzed variants would be extremely beneficial to improve reader’s understanding.
  6. Table 5: why the specific values for organic matter pollution (BOD, COD) are so low when compared to nutrients? This is another key point to be clarified.
  7. Table 6-Fig. 2: COD is written incorrectly.
  8. Fig. 2 is redundant; decide if keeping Table 6 or Fig. 2, as the information that they are presenting is exactly the same. More generally, try not to replicate in Figures the data that are already presented in Tables and vice versa.
  9. Table 7: why pollutant removal is slightly increasing over time?
  10. At present, the distinction between Materials and Methods and Results and discussion sections is not clear/correct: in the former, the authors should recap the applied methodology and the main tested hypothesis, while in the latter all the results should be presented, including discussion with relevant literature in the field.
  11. Results and discussion: the discussion of present results with other literature studies is practically absent; the authors should try to improve the discussion.
  12. Conclusions sections is too long; it could be shortened at least by 20%.
  13. English language should be refined: native speaker revision would be beneficial.
  14. Some recent literature studies, focused on economic sustainability of WWTPs, could help increase the discussion: e.g., see 10.1051/matecconf/201927606019, 10.3390/su13020982 and 10.3390/w12102926.

Author Response

Dear Editor and Reviewer 1,

We are very thankful to you for professional organization and quick review process. We are especially grateful for the constructive comments and suggestions of reviewers, which helped us a lot in improving the quality of our paper. The paper has been revised carefully based on the comments from the all reviewers and we have marked the changes by using the ‚‚Track changes“ in the revised paper.

We would like to express our sincere gratitude to the Reviewer 1 for your positive comment. We completely respect your opinion on the working version of our paper. Based on reviewers comments the entire paper has undergone significant changes, we have accepted all the comments and significantly improved the quality of our manuscript. We sincerely hope that the revised version of the manuscript will deserve your positive opinion and recommendation of the paper for publication in the reputable journal Water.

Reviewer 2 Report

I am glad to have the opportunity to review the manuscript entitled "

Author Response

Dear Editor and Reviewer 2,

We are very thankful to you for professional organization and quick review process. We are especially grateful for the constructive comments and suggestions of reviewers, which helped us a lot in improving the quality of our paper. The paper has been revised carefully based on the comments from the all reviewers and we have marked the changes by using the ‚‚Track changes“ in the revised paper.

We would like to express our sincere gratitude to the Reviewer 2 for your positive comment. We completely respect your opinion on the working version of our paper. Based on reviewers' comments the entire paper has undergone changes, we have accepted all the comments and significantly improved the quality of our manuscript. We sincerely hope that the revised version of the manuscript will deserve your positive opinion and recommendation of the paper for publication in the reputable journal Water.

Round 2

Reviewer 1 Report

The authors answered in a good way to all reviewer observations and comments. I consequently recommend manuscript acceptance in its present form.

Author Response

We are very thankful to you for professional organization and quick review process

Reviewer 2 Report

I would like to appreciate the authors for addressing my concerns.

Good Luck 

Author Response

(The authors gave the same response as above.)
